# Variations in the Reproductive Strategies of Different *Charadrius alexandrinus* Populations in Xinjiang, China

**DOI:** 10.3390/ani13142260

**Published:** 2023-07-10

**Authors:** Peng Ding, Zitan Song, Yang Liu, Tamás Székely, Lei Shi, Mardan Aghabey Turghan

**Affiliations:** 1College of Animal Sciences, Xinjiang Agricultural University, Urumqi 830052, China; dpengde@126.com; 2State Key Laboratory of Biocontrol, School of Ecology, Sun Yat-Sen University, Guangzhou 510275, China; 13624326602@163.com (Z.S.); liuy353@mail.sysu.edu.cn (Y.L.); 3Milner Centre for Evolution, Department of Biology and Biochemistry, University of Bath, Bath BA1 7AY, UK; bssts@bath.ac.uk; 4Department of Evolutionary Zoology, University of Debrecen, 4032 Debrecen, Hungary; 5State Key Laboratory of Oasis and Desert Ecology, Xinjiang Institute of Ecology and Geography, Chinese Academy of Sciences, Urumqi 830011, China

**Keywords:** Kentish plover, life history, reproductive output, breeding behavior, inland arid area

## Abstract

**Simple Summary:**

In response to the harsh natural environment in the arid lands of Xinjiang, China, Kentish plover *Charadrius alexandrinus* populations in different regions show flexibility in their breeding strategies to cope with the changeable environment. Of our three study areas, one (Taitema Lake) is distinctly characterized by barren terrains and a harsh climate. Here, compared with the two other study areas, the tarsometatarsi of the female plovers were shorter but their egg size and clutch volume were significantly larger than in the two other populations. With the postponement of laying dates, partly due to phenological differences, the two other populations, respectively, show the characteristics of a small clutch size and a relatively small egg size and a decrease in reproductive output. In summary, our data support the hypothesis that Kentish plover populations show flexibility in their breeding strategies to cope with the harsh natural environment in the arid lands of Xinjiang, China. The relatively high average clutch size and average egg size observed in the study area also show that the saline wetlands in Western China are an important breeding habitat for Kentish plover. Therefore, in addition to the limitation of the birth cloaca, the explanation of the egg length allometry may be related to other factors, such as nest size, structure, shape, climate, and brood desertion, etc. Future research will be recommended to further clarify the underlying mechanism of the species’ reproductive strategies in response to regional environmental modification resulting from anthropogenic landscape features (human activities) and global climate change.

**Abstract:**

Due to the influence of bio-geographical and environmental factors, as well as anthropogenic landscape features, organisms show different reproductive strategies among different populations. There is a lack of detailed information on the reproductive biology of Kentish plover *Charadrius alexandrinus* in arid lands in Central Asia. In this study, we summarized the characteristics of the reproductive biology of three geographically distinct plover populations in Aibi Lake in northwestern Xinjiang, Taitema Lake in southern Xinjiang and artificial reservoirs around Urumqi City in northern Xinjiang, based on 440 eggs from 158 nests observed and analyzed from April to July of 2019 and 2020. We found that there was no significant difference in clutch size among the three populations. However, the egg size of the Taitema Lake population was significantly larger than those of the other two populations, whilst the egg volume and clutch volume of the artificial reservoirs’ populations were significantly larger than that of Aibi Lake. With the postponement of laying dates, the northern and northwestern populations showed the characteristics of a small clutch size and a relatively small egg size, respectively, and a decrease in reproductive output. The heavier female plovers in Taitema Lake laid eggs earlier, and there was a significantly positive correlation between female body mass and clutch size and egg size. The tarsometatarsus length of the female plovers was significantly positively correlated with the reproductive output in all three populations. The model selection results show that female body size and ambient temperature restrict the egg size and reproductive output of Kentish plovers, which is consistent with the upper limit hypothesis of the maternal condition and maternal constraint. Our data support the hypothesis that Kentish plovers show distinct flexibility in their breeding strategies to cope with the harsh natural environment in the arid lands of Xinjiang, China. The results of a relatively high average clutch size and average egg size imply that the saline wetlands of Western China are important breeding habitats for Kentish plovers.

## 1. Introduction

Reproductive life history represents the overall trade-off between the investment in eggs and clutch and the timing of reproduction, important events in the reproductive process of biological organisms, with a particular focus on the mechanism of why different species in the same environment or the same species in different environments may develop different reproductive strategies [1,2,3,4,5,6,7,8,9,10,11]. Reproduction, the content of greatest concern to ornithologists, is the fundamental process for life history, and reproductive strategy is the decisive factor that affects the fitness of animals, directly through individual survival and reproduction, population dynamics, and persistence [3,10,12,13,14,15,16]. A large number of studies have described the diversity of reproductive life history strategies among birds due to variations in environmental conditions and individual morphology [16,17,18,19,20,21,22,23,24,25,26,27,28]. Additionally, the body size and the reproductive characteristics in populations of the same species are also influenced by environmental and phylogenetic factors [19,22,29]. The reproductive life history characteristics of birds, such as egg-laying time, egg size, clutch size, and clutch volume, are usually considered to be the main indicators to measure reproductive output [10,17,19,22,27,28,29,30]. These indicators can effectively reveal the different reproductive traits of birds in different environments due to adaptive trade-offs and constraint of physiological mechanisms, which can be interpreted as different reproductive life history strategies [17,19,22,27,28,29,30,31]. For example, the egg size and clutch size of birds can be adjusted according to changes in the egg-laying date, ambient temperature, food resources, and female physical condition [4,17,32,33].

Life history theory seeks to explain how natural selection and other evolutionary forces shape organisms to optimize their survival and reproduction in the face of ecological challenges posed by the environment [1,2]. The trade-off between egg size and clutch size is one of the core principles of life history evolution, in which numerous studies have reported a rather positive covariance between egg size and clutch size: good-quality females may lay more and larger eggs compared to low-quality females [1,33,34,35], and adequate food resources usually lead to the early laying of eggs and larger clutch sizes instead of larger eggs [6,36,37]. Theoretically, the reproductive output of parent birds that lay eggs at a fixed number can be increased by increasing the size of a single egg, but the physiological constraint hypothesis predicts that the reproductive output will be constrained by the maternal condition, which is specifically manifested in larger individuals laying larger eggs, while smaller individuals lay smaller eggs due to insufficient energy storage in their bodies [38]. Body size can partly represent the female’s energy storage, so as their body mass increases, the female can supply more energy (egg mass) or lay more eggs [39]. The size of an egg has costs and benefits, and individuals might balance these costs and benefits when allocating resources to the size of the egg [40]. According to life history theory, females should have some flexibility in the allocation of resources for eggs, including laying optimally sized eggs and small clutch sizes in the case of abundant resources and laying a relatively small-sized egg size but a large clutch size in the case of limited resources [2,28,41,42]. However, a constant number of offspring has been found in some groups. Birds provide care to their offspring during laying, incubating, and brooding. As the incubation capacity hypothesis predicts that the maximum clutch size that females can lay is constrained by the incubating capacity (e.g., incubating spots size), this may explain why most species of shorebirds have a relatively invariant clutch size [43,44]. Thus, most plovers vary their reproductive output by varying their egg size rather than their clutch size [6].

On the other hand, the morphological constraint hypothesis holds that an egg’s length or width is constrained by a female’s partial morphological features. When the egg width is constrained, females can increase their energy investment by laying longer eggs, for example, lizards can lay longer eggs to increase their energy supply to eggs as the tail base width of female lizards constrains the egg width [45]. In light of this phenomenon, some scholars have put forward the concept of egg-shape allometry, which states that with the increase in egg size, the growth rate of the egg length and width is different, and the pattern of allometry in egg shape can be checked by comparing the regression slope between egg length and egg width [46]. The upper limit hypothesis of maternal constraint predicts that the egg width is physically constrained by the cloaca of the bird, whilst the increase in the egg size is mainly achieved by increasing the egg length, which is manifested as the egg length allometry [42,46].

The Kentish plover *Charadrius alexandrinus* is a small shorebird (with a body length of 14~17.5 cm) with an extremely large range and multiple geographical races, with the bird living in both temperate and subtropical climate zones on four continents. Kentish plovers often inhabit open and flat coastal beaches and bare land around saltwater lakes and inland lakes [47,48]. The breeding period of Kentish plover is usually from March to August every year with certain variations in different regions. They usually lay three oval eggs per nest, but individual females may produce two or four egg clutches with a length of 29~35 mm, a width of 22~25 cm, and a mass of 8~11 g [9,47,49]. Female birds lay eggs at 2-day intervals until incubation starts after the last egg has been laid. The incubation period is approximately 23 to 29 days [9,50]. In China, Kentish plovers breed in coastal areas, inland lakes, and near reservoirs in the north, inhabit all provinces during migration, and can be seen throughout most of the southeastern coast during non-breeding periods [51,52]. Because of their wide distribution, the breeding strategies of Kentish plovers are diverse and have strong plasticity, which attracts the attention of scholars in animal ecology, behavior, and evolution [43]. Variations in breeding strategies are common phenomena in birds living in different distribution ranges [10,43,53,54]. For example, reproductive characteristics, such as egg length and width, clutch size, laying date, incubation rate, growth rate, brood desertion, and adult survival, often differ distinctively between northern and southern bird species [7,8,55]. Several studies have investigated how reproductive traits vary within a single species as the elevation increases and the climate becomes more severe [21,56,57,58]. We hypothesize that, as far as the avian life history variation along altitudinal gradients is concerned, breeding in high-elevation habitats results in a shift to a shorter life history strategy within a single species.

At present, the reproductive biology of the Kentish plover has been studied mainly in coastal areas, and most of the data come from Europe and North Africa [44,47,59,60]. In China, the study of Kentish plover reproductive biology mainly focuses on the populations in Bohai Bay [61] and Qinghai Lake [62]. The populations in the inland arid areas of Western China have drawn less attention [63]. Due to the influence of geography, temperature, and the environment, organisms show different reproductive strategies among different populations [42,44,62,64]. In view of the harsh natural environment of the inland arid areas of Xinjiang, we hypothesize that Kentish plovers may show flexibility in their breeding strategies to cope with the changeable environment. Here, we focus on the reproductive traits of the Kentish plover to reveal the reproductive strategy of this bird adapting to different localities in the arid desert areas of Xinjiang (Figure 1), where precipitation varies greatly but the temperature differences are not significant (Figure 2). During the early stage of the breeding season, the climate of Taitema Lake is extremely dry with frequent dust storms. Hence, we hypothesize that the plovers of Taitema Lake adapt to the poor climate conditions by laying larger eggs and shortening their incubation period. According to the physiological constraint hypothesis, we hypothesize that female traits such as body mass or tarsometatarsus length determine reproductive output, which is manifested in different egg sizes and clutch sizes in different populations. According to the upper limit hypothesis, we hypothesize that larger eggs show a more positive allometry in egg shape. Therefore, the primary objective of this study was to provide baseline information about the breeding strategies of Kentish plover populations in the inner arid regions of Xinjiang, one of the important breeding areas for this species in China.

## 2. Materials and Methods

### 2.1. Study Area

The current study was conducted, during the breeding season of Kentish plover, from April to July of 2019 and 2020. We collected and compared a variety of reproductive traits among three geographically distinct Kentish plover populations of Xinjiang, including an Aibi Lake (AL) population in northwestern Xinjiang, a Taitema Lake (TL) population in southern Xinjiang, and an artificial reservoir (AR) group population around Urumqi City in northern Xinjiang (Figure 1).

The Aibi Lake Basin is a closed basin located in the inland area of the Junggar Basin in Xinjiang in northwestern China (43°380–45°520 N, 79°530–85°020 E, with an average altitude of 200 m) [65]. It has a total area of 50,621 km^2^ where plains make up 25,762 km^2^. Aibi Lake, one of the important breeding habitats and temporary rest stations for migratory birds in western China, is under the jurisdiction of the Xinjiang Aibi Lake Wetland National Nature Reserve. The lake, characterized by its dry and hot weather, high temperatures, low precipitation, and little human disturbance, is the largest Salt Lake in western China, with a water area of 520 km^2^. The average annual temperature is 8.3 °C, and the average annual precipitation is 90.9 mm. The average annual precipitation on the surface of the lake is about 95 mm, with the annual evaporation able to reach as high as 1315 mm. Strong winds with a maximum wind speed of 55 m/s are more likely to occur from April to June. The Kentish plover populations there usually nest on saline alkali land, sand land, and gravel substrates in the region [66].

The reservoir group including Liuchengzi Reservoir and Wushihua Reservoir around Urumqi City is located at the north edge of Bogeda Mountain. They are typical plain reservoirs mainly used for aquaculture and irrigation. The specific study sites are located on the southeast bank of Liuchengzi Reservoir (N 44.255835°, E 87.885218°, with an average altitude of 480 m) and the east of Wushihua Reservoir (N 44.196667°, E 87.741022°, with an altitude of about 475 m). Since the distance between the two sites is only 13 km, and there is no difference in female morphology and egg characteristics between the populations from these two sites, we combined those two populations as an artificial reservoir population. This area is close to villages and towns, mostly surrounded by reclaimed farmland or industrial parks with strong human disturbance, such as livestock and road construction, which accidentally cause the failure of nests.

Taitema Lake (TL), with a water surface of 300 km^2^, is located in the southeast margin of Taklimakan Desert, about 50 km north of Ruoqiang County, Bayingolin Mongol Autonomous Prefecture. It is the terminal lake of three river systems: Tarim River, Cherchen River, and the rivers on the northern slope of the Altun Mountains [62,67]. The Taitema Lake region has an extreme continental climate, making it extremely arid. The average annual precipitation is 17.4–42 mm, the average annual evaporation is 2500–3000 mm, and the extreme maximum temperature is 43.6 °C. Furthermore, above-ground vegetation is sparse. The study area is located on the southeastern shore of the lake (N 39.412433°, E 88.517855°, with an average altitude of 800 m). The breeding habitat of plovers is basically a harsh desert composed of desert grassland, sand land, and sand dunes, which is well-preserved with little human disturbance [67,68].

### 2.2. Data Source

In this study, the regional meteorological data (Figure 2) came from the National Meteorological Information Center (https://www.nmic.gov.cn/en/, accessed on 16 August 2021) of the China Meteorological Administration, with data on temperature and precipitation obtained from the weather station, closest to the study region. We calculated the average value every ten days, including the daily average temperature (AT), the daily maximum temperature (DT_max_), the daily minimum temperature (DT_min_), the daily temperature difference (DD), and the sum of rainfall (RF) during the breeding season from 27 March to 24 July each year in 2019 and 2020.

### 2.3. Data Collection and Traits Measurement

We found the nests by watching the birds with a telescope and binoculars, and by searching on foot inside the nesting grounds during the breeding season [69]. We monitored the nests every 3–4 days each year during the breeding season until the hatching of the eggs or the failure of the breeding attempt. The morphological traits of the female parent and their eggs’ size characteristics (length and width) were measured with a digital caliper (0.01 mm) and then weighed with a digital scale (0.01 g) [69]. GPS data of the nests were recorded. We captured the breeding female parent with a walk-in funnel trap placed over the nests that had been incubated for at least seven days [62]. The 12, 30, and 42 female plovers were captured and measured at the AL, AR, and TL locations, and the number of nests was 20, 77, and 61 for AL, AR, and TL, respectively. The same combination of the number of recorded nests (20, 77, and 61 at the AL, AR, and TL locations, respectively) also included the abandoned or trampled nests. The captive parent were then banded with a metal ring and three colorful rings with unique combinations as the marker. Female body mass (FBM, g) is considered as the female’s weight after laying all eggs. Female tarsometatarsus length (FTL, mm) is the tarsometatarsus length on the right side of the female. Egg mass (OEM, single egg weight, g), clutch size (CS, the total number of eggs laid in a single breeding period), egg length (EL, linear length at the longest end of the egg, mm), egg width (EW, linear length at the maximum width of the egg, mm), and egg shape (ES, ratio of egg width and egg length) were measured to analyze the allometry in egg shape.

We also calculated egg volume (EV, cm^3^): *EV* = *Kv* × *L* × *W*^2^, where *Kv* = 0.5236 − (0.5236 × 2 × (*L*/*W*)/100), *L* = egg length, *W* = egg width, and clutch volume (CV, the sum of the volumes of all eggs in a single breeding period, cm^3^) [70]. We estimated the laying date (LD) through observation and floating the eggs in lukewarm water [69]. We used the Julian day, which we calculated as the number of days between the 1 April (i.e., 1) of each year, as the egg-laying date. The time interval between the beginning and the end of the hatching of the plover was taken as the incubation period (IP). The monitoring of the AL population was too difficult and the number of nests was too small, so the data of the incubation period could not be obtained through continuous monitoring.

### 2.4. Statistical Analysis

The collected data were sorted through with Excel data. Non-normally distributed data were log10 converted to meet the assumption, i.e., the Kolmogorov–Smirnov normality test and Levene’s variance homogeneity test. We used one-way ANOVA to make multiple comparisons on the female body mass, tarsometatarsus length, egg size, clutch volume, and other traits of the three populations with post hoc Tukey’s tests. For the data that did not meet the above assumptions, we used the nonparametric Kruskal–Wallis test to examine data differences between the three populations with post hoc Steel Dwass tests (multiple comparisons). We used analysis of an independent *t*-test to examine the annual differences in the reproductive traits of each population; the results showed no significant annual differences in reproductive traits among them, so two years of reproductive traits data from each population were combined for subsequent analysis. Model II regression in the “lmodel2” package was used to carry out regression analyses between the egg size, clutch size, clutch volume, and laying time one by one, as well as between the egg size, clutch size, clutch volume, body mass, and tarsometatarsus length of their female parent one by one, so as to verify the effects of laying time and female morphology on egg size and clutch size.

For the egg shape allometry, we first determined the regression between egg length and egg width. If the slope was greater than 1, there existed egg shape allometry. Then, residuals regarding the egg length and egg width with clutch size were extracted, respectively, to eliminate the effects of the clutch size on the egg length and width [42], and regression was determined between the residual egg length and the residual egg width to test whether there is allometry for the egg shape after removing the effects of the clutch size.

Model selection analysis was conducted to test the effect of environmental factors and female morphological traits on egg volume and clutch volume. Daily average temperature, daily temperature difference (DD), daily maximum temperature, daily rainfall, laying date, clutch size, female body mass, and female tarsometatarsus length were taken as independent variables to carry out the multinomial logistic model analysis. The populations were included as a fixed index, and the parameter years were entered as a random effect in order to control for interpopulation phylogenetic relatedness.

Akaike information criterion (*AIC*) was used to compare models and determine two principles of the best model screening: (1) the minimum *AIC* value is required and (2) the dealt value of the model < 2. All data are listed in the form of the mean ± standard error (mean ± SE), and all operations were completed in R.v.3.6.3 software (R Development Core Team, https://cran.r-project.org/), using the packages “ggplot” and “gplots”. Differences were considered significant when *p* < 0.05.

## 3. Results

### 3.1. Female Reproductive Traits among the Populations

A total of 440 eggs from 158 nests of Kentish plovers in three geographically distinct populations were measured from April to July of 2019 and 2020. Among them, there are 54 eggs from 20 nests in AL, 175 from 61 nests in TL, and 211 from 77 nests in the AR.

The results show that there is no significant difference in female body mass among the three populations, while the female tarsometatarsus length of the AL population was significantly longer than that of the other two populations. There was also no significant difference in clutch size among the populations, but there were significant differences in egg volume and clutch volume in the TL population, which were larger than those of the other two populations. Additionally, the egg width, egg volume, and clutch volume of the AR population were significantly larger than those of the AL population (Table 1).

### 3.2. Laying Date and Reproductive Traits

According to our observations, the AR population is strongly affected by the interference of artificial water level control and regional grazing. The first laying peak appears at the early stage of the breeding season, while the second laying peak immediately follows the expansion of the suitable nesting area in early June when the water level retreats to its lowest point. In addition, the average incubation period (IP) of the plover population in the AR is significantly longer than that of the TL population (Table 1).

Among the three populations, there was a significant negative correlation between the female body mass and the laying date of the plover in TL (Figure 3A), which showed that the heavier females laid eggs earlier. There was also a significant negative correlation between the laying date and clutch size and clutch volume of the AL population (Figure 3B,F). The laying date of the AR population significantly restricted the egg volume (Figure 3C–E) and clutch volume (Figure 3F). With the postponement of the laying date, the plovers there produce relatively small eggs, representing lower reproductive output.

### 3.3. Reproductive Output and Female Traits

The results show that heavier and larger females lay larger eggs. Among the three populations, the female body mass in TL was significantly positively correlated with clutch size (Figure 4A), egg width, egg volume (Figure 4B,C), and clutch volume (Figure 4D). There was an extremely significant positive correlation between the female tarsometatarsus length and egg length (Figure 5A) and clutch volume (Figure 5B) in the AR population.

### 3.4. Model Selection

The egg size and clutch volume model revealed that that body mass is again a significant predictor of variation in egg size. The results of the egg size model selection show that the best model for egg volume of the three populations of birds includes female body mass, female tarsometatarsus length, and average temperature. The *AICc* value of the model is −197.50, and the weight is 0.948 (Table 2). The best model of the clutch volume of three populations of birds includes female body mass, egg length, and average temperature. The *AICc* value of the model is −197.00, and the weight is 0.611 (Table 2).

### 3.5. Allometry

The regression slope (slope = 1.625, Figure 6A) between the egg length and egg width was significantly larger than 1 (*p* < 0.0001), indicating that there was allometry in the egg shape, and the growth rate of the egg length was significantly faster than that of the egg width. After extracting the regression residuals of the egg length, egg width, and clutch size, the regression slope between the egg length residuals and egg width residuals (slope = 2.277, Figure 6B) was also significantly larger than 1 (*p* < 0.0001), indicating that after eliminating the effect of clutch size, the egg shape allometry still exists, and the growth rate of the egg length is significantly faster than that of egg width.

## 4. Discussion

### 4.1. Differences in Reproductive Traits among the Populations

The average clutch size in the three populations (2.86 ± 0.35~2.96 ± 0.35) was similar to that at other sites [44,47,48,49,50]. The proportion of nests with three eggs in the three populations in the study area was higher than 85%, which is higher than about 70% reported at other sites [9,47]. Compared with other sites, the average egg volume was also larger [47,69,71]. The clutch size and the egg volume are generally considered to be related to food supply [37]. Larger eggs have been proven to improve the hatching success rate and increase the initial energy reserve of nestling [36,40]. For example, the TL population in a relatively harsh environment with high daily average temperatures produces larger eggs relative to body size, which strongly supports our hypothesis that the plovers of TL adapt to the poor climate conditions by laying larger eggs and shortening their incubation period.

### 4.2. Relationship between Reproductive Traits and Laying Date

All three populations began to incubate in April, which was later than that in European and Northern African populations [47,60]. The shorter breeding season may be related to the late start of laying, which may be limited by climatic conditions (mainly low temperatures in early spring) [72]. Secondly, the laying data are life history characteristics, which depend on the habitat conditions, including altitudinal gradients. Compared with other populations, the TL population lives under poor early climate conditions due to the higher altitude (850 m), which results in shorter female tarsometatarsus length, later laying dates, and shorter incubation periods, resulting in the shortening of the breeding season in TL [63], which strongly supports our hypothesis that breeding in high-elevation habitats results in a shift to a shorter life history strategy within a single species. Since the interference of livestock in the late breeding season is the main reason for the nesting failure of the AR population (unpublished data), laying begins earlier, which may be affected by temperature, resulting in a long incubation period. The results of another of our manuscripts in preparation indicated that there were differences in the nest survival rates among the populations, in which the rate of the TL population was the highest (0.702, n = 61 nests) and that of the AR population was the lowest (0.296, n = 77 nests) due to predation, parental desertion, and the higher possibility of being trampled by livestock. We also discovered that the nest survival rate of the AR population decreased with the postponement of the breeding season, and the females there would lay relatively small eggs, resulting in lower reproductive output. These results testified the fact that the AR population manifested relatively flexible reproductive strategies, such as a trade-off between egg size and clutch size, in response to the lower survival rate. Additionally, higher levels of human interference, such as artificial water level control, regional grazing, road construction, as well as predation risks also affect the end of reproduction by causing birds to leave early, which led to higher nesting failure in the AR population. The later start of laying and shorter laying time of the TL population may be the adaptation strategy of the population to severe environmental conditions, which is consistent with the reproductive limit hypothesis at high altitudes [63,73].

There was a significant negative correlation between the laying date and egg volume and clutch volume of the birds in the AR, which showed that the egg size and clutch volume decreased with the passage of the laying date. There may be four reasons for this phenomenon. Firstly, individuals who lay eggs early may be older females. They arrive early and start breeding early. Usually, young females tend to lay eggs late, and lay smaller eggs [6]. Secondly, according to the embryonic temperature hypothesis [73], larger eggs can obviously reduce the mortality of offspring in cold environments [74]. At the early stage of reproduction, the temperature is low, and it is beneficial for hatching to produce larger eggs. Thirdly, it is also possible that the eggs produced in the later stage belong to the second clutch, which is due to compensatory laying after the failure of hatching in the first clutch. Usually the eggs in the second clutch are smaller. Finally, in the inland arid areas where water resources are relatively scarce, the change in water level directly affects the potential habitat for nesting. For example, after the water level retreats due to artificial control in the middle and late breeding period, a large area of ideal nesting areas is created, resulting in explosive nesting and laying in the AR.

### 4.3. Egg Size and Reproductive Output Are Affected by Both Maternal and Environmental Conditions

There was a significant positive correlation between the clutch size, egg size, clutch volume, and bird postpartum weight in TL. In the face of the local harsh climate, larger females will produce a relatively large number of eggs to improve the hatching success rate and the initial energy reserve of nestling [36,49,75]. It has been suggested that egg size has a significant impact on the performance of nestling [40]. The tarsometatarsus length is an important identification indicator of bird age or quality. Our results show that there is a significant positive correlation between reproductive output and the female tarsometatarsus length in the AR. In addition, food supplementation usually leads to early laying and a larger clutch size rather than larger eggs, which is common in birds [36,37]. However, we noticed that the females at the AL location, which had significantly longer tarsometatarsus lengths relative to the females at the AR and TL locations, did not have greater reproductive output. This result suggests that other environmental factors at this location have a stronger influence that negates the normal benefit associated with increased tarsometatarsus length and this would need to be delineated in subsequent research.

As the parent birds lay eggs at a fixed number, Kentish plover can increase their reproductive output by increasing the size of a single egg, but this is limited by maternal and environmental conditions. According to the model selection results (Table 2), the egg volume and clutch volume of the three plover populations are affected by the females’ traits, including the females’ body mass and the females’ tarsometatarsus length, which is consistent with our hypothesis that female traits, such as body mass and tarsometatarsus length, determine reproductive output, which is manifested in different egg sizes and clutch sizes in different populations. The reproductive output of individuals with larger body sizes is also greater, which conforms to the prediction of the physiological constraint hypothesis [38]. Additionally, the clutch volume of the plover is also affected by the egg length, which may be due to the limitation of its hard shell by the birth cloaca, and the size of a single egg can only be increased by increasing the egg length, which supports the upper limit hypothesis of maternal constraint [42,46].

### 4.4. The Allometry of Egg Shape

Egg shape, the same as egg size, is a highly variable characteristic in the life history of birds [2,31,48,76]. However, little is known about the adaptive significance of bird egg shape and how the differences arise and evolve [77]. We found that the growth rate of the egg length of Kentish plover is significantly faster than that of egg width (Figure 6), which confirms the prediction of the upper limit hypothesis and is consistent with the results of the model selection. Due to the intermittent laying mode of birds (such as laying every other day), only one egg is laid at a time. Theoretically, the size of a single egg will not be limited by the volume of the abdominal cavity. At the same time, the egg width of birds will not be limited due to the open pelvis (some birds lay nearly round eggs). Therefore, in addition to the limitation of the birth cloaca, the explanation of the egg length allometry may be related to other factors, such as nest size, structure, shape, and the climate [78]. Further research is needed to determine the underlying mechanism.

## 5. Conclusions

The present study strongly supported our hypothesis that Kentish plover may show flexibility in breeding strategies to cope with the changeable environment. The three populations of plovers have precocial chicks and invariant clutch sizes. The laying date of the northern Xinjiang populations was earlier than that of the southern Xinjiang populations. The environment in TL is the most barren and the climate is the harshest among the three, and the female tarsometatarsus length is the smallest, but the egg size and the clutch volume are significantly larger than those of the AR and AL populations. Female body mass has a significant positive correlation with egg size, clutch size, and clutch volume in all three populations. The egg width, egg volume, and clutch volume of the AR population were significantly larger than those of the AL population, and the female tarsometatarsus length of the population has a significant positive correlation with the egg length and clutch volume. The single egg size and clutch volume of the three bird populations are affected by morphological constraints and maternal constraints. In total, our observation results of a relatively high average egg size and clutch size imply that the saline wetlands of Western China are an important breeding habitat for Kentish plover. Finally, future research is recommended to further clarify the underlying mechanism of the reproductive strategies of the species in response to regional environmental modification resulting from anthropogenic landscape features (human activities), altitudinal gradients, and global climate change.

## Figures and Tables

**Figure 1 animals-13-02260-f001:**
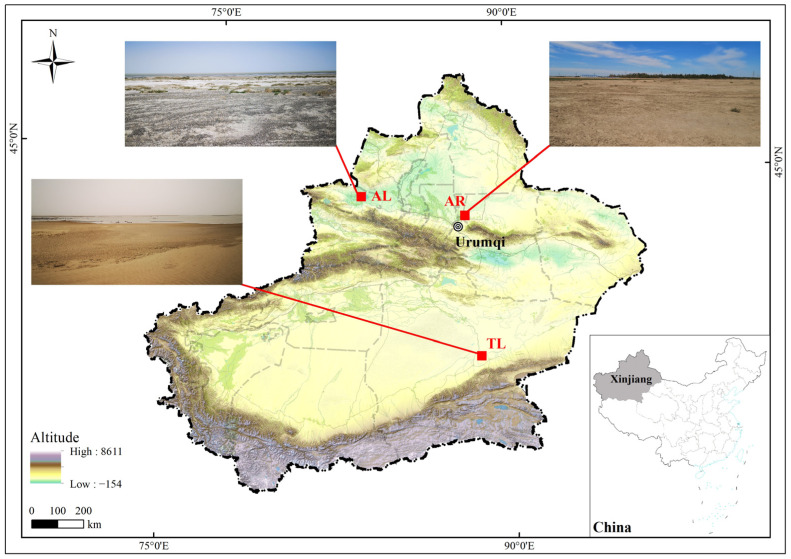
Map of the study areas of three populations of *Charadrius alexandrinus* in Xinjiang.

**Figure 2 animals-13-02260-f002:**
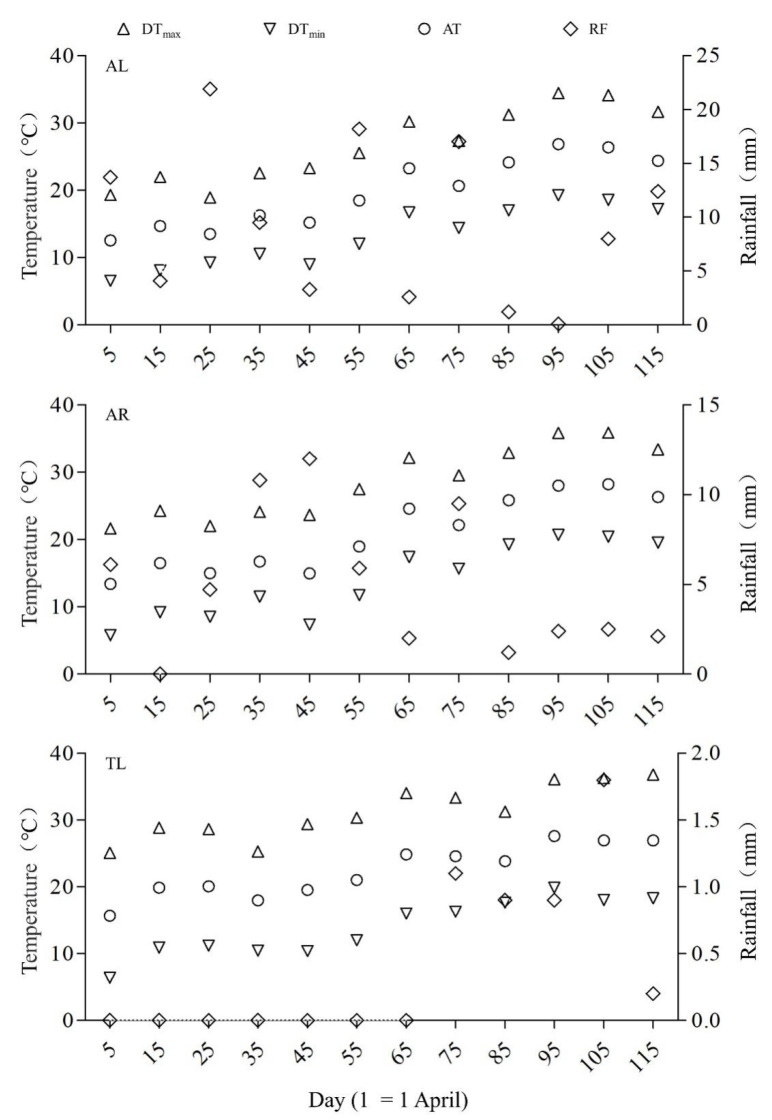
Temperature and precipitation in the three regions (Including the daily average temperature (AT, ○), the daily maximum temperature (Dt_max_, △), the daily minimum temperature (DT_min_, ▽), and the sum of rainfall (RF, ◇). Data were averages for 2019 and 2020.).

**Figure 3 animals-13-02260-f003:**
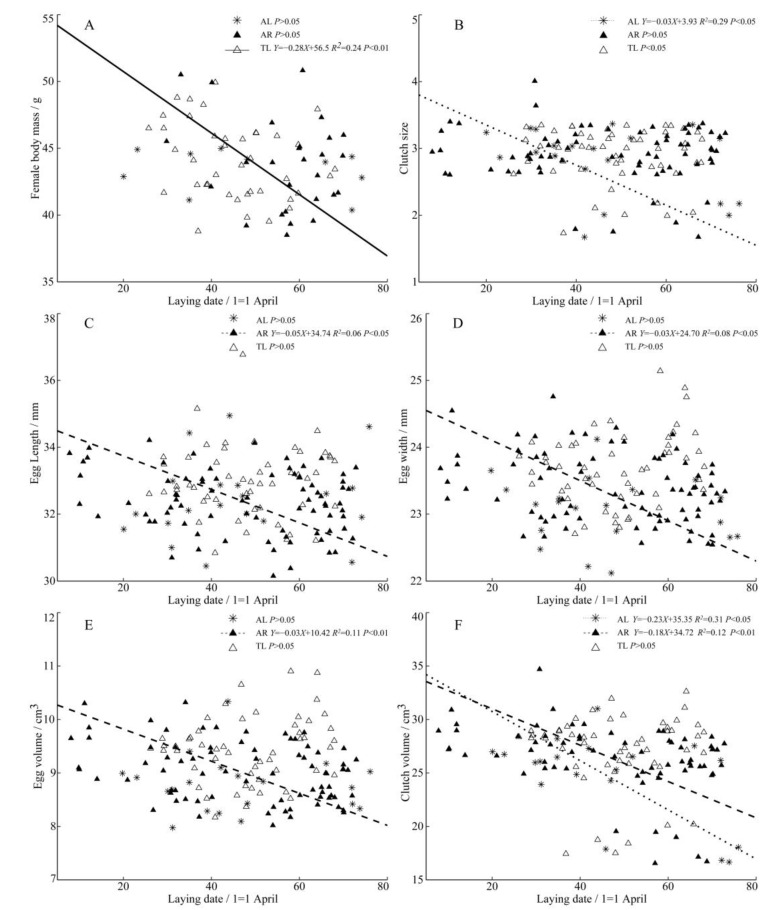
Regressions of female body mass (**A**), clutch size (**B**), egg length (**C**), egg width (**D**), egg volume (**E**), clutch volume (**F**) and laying date from three populations of *C. alexandrinus*. Fitted reduced major axis regression model and statistical significance (*p* < 0.05) are indicated in each case. AL, Aibi Lake–Asterisk; AR, Artificial Reservoir–Black Triangle; TL, Taitema Lake–White Triangle. Points were jittered using the geom jitter function.

**Figure 4 animals-13-02260-f004:**
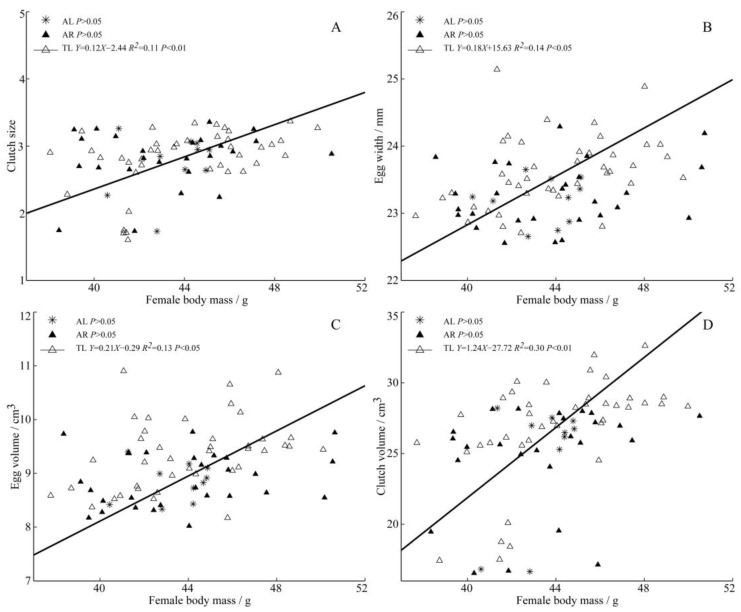
Regressions of female body mass and clutch size (**A**), egg width (**B**), egg volume (**C**), clutch volume (**D**) from three populations of *C. alexandrinus*. Fitted reduced major axis regression model and statistical significance (*p* < 0.05) are indicated in each case. AL, Aibi Lake–Asterisk; AR, Artificial Reservoir–Black Triangle; TL, Taitema Lake–White Triangle. Points were jittered using the geom jitter function.

**Figure 5 animals-13-02260-f005:**
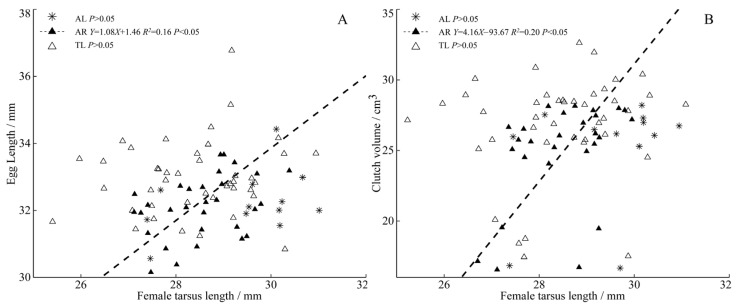
Regressions of female tarsometatarsus length and egg length (**A**), clutch volume (**B**) from three populations of *C. alexandrinus*. Fitted reduced major axis regression model and statistical significance (*p* < 0.05) are indicated in each case. AL, Aibi Lake–Asterisk; AR, Artificial Reservoir–Black Triangle; TL, Taitema Lake–White Triangle. Points were jittered using the geom jitter function.

**Figure 6 animals-13-02260-f006:**
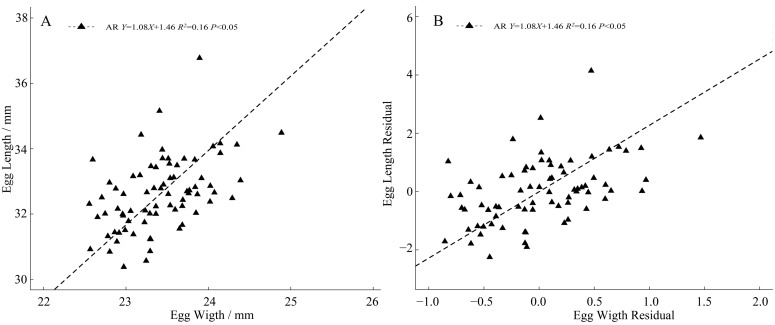
Allometry of egg shape of *C. alexandrinus* ((**A**) egg shape allomery, (**B**) egg shape allometry by residuals). The eggs–Black Triangle; Regression relationship between the egg length and egg width−Dashed Line.

**Table 1 animals-13-02260-t001:** Descriptive statistics of female reproductive traits in three populations of *C. alexandrinus*.

Traits	AL (*N* = 20)	AR (*N* = 77)	TL (*N* = 61)	*F*-Level	*p*-Value
Female body mass *#, g	43.42 ± 1.52	43.79 ± 3.37	43.95 ± 2.93	*F*_2, 75_ = 0.127,	*p* = 0.881
Female tarsometatarsus length *#, mm	29.47 ± 1.12 ^a^	28.51 ± 0.84 ^b^	28.38 ± 1.23 ^b^	*F*_2, 81_ = 4.602,	*p* = 0.012
Egg length †, mm	32.38 ± 1.30 ^B^	32.36 ± 1.08 ^B^	33.01 ± 1.24 ^A^	*F*_2, 437_ = 15.810,	*p* < 0.001
Egg width #, mm	23.10 ± 0.55 ^Bb^	23.38 ± 0.55 ^Ba^	23.64 ± 0.54 ^A^	*F*_2, 436_ = 23.070,	*p* < 0.001
Egg shape #	0.71 ± 0.03	0.72 ± 0.03	0.72 ± 0.03	*F*_2, 436_ = 3.961	*p* = 0.020
Egg volume #, cm^3^	8.80 ± 0.60 ^Bb^	9.02 ± 0.58 ^Ba^	9.40 ± 0.64 ^A^	*F*_2, 436_ = 28.790	*p* < 0.001
Clutch size †	2.75 ± 0.43	2.96 ± 0.31	2.87 ± 0.34	*χ*^2^ = 5.929	*p* = 0.052
Clutch volume †, cm^3^	24.57 ± 4.02 ^Bb^	26.39 ± 3.19 ^a^	27.07 ± 3.50 ^A^	*χ*^2^ = 12.249	*p* = 0.002
Incubation period &, day	/	26 ± 1.33 ^a^	25.09 ± 0.51 ^b^	*U* = 54.50	*p* = 0.037

Note: Different capital letters indicate significance at the *p* < 0.01 level; Different lowercase letters indicate significance at the *p* < 0.05 level. The monitoring of AL population is too difficult and the number of nests was too small, so the data of incubation period could not be obtained by continuous monitoring. * AL n = 12; AR n = 30; TL n = 42. # ANOVA; † Nonparametric Kruskal–Wallis test; & Nonparametric Mann–Whitney U test.

**Table 2 animals-13-02260-t002:** Model selection for reproductive output of *C. alexandrinus*.

Parameter	Optimization Model	
	Egg Volume	Clutch Volume
Clutch size		
Female body mass	+	+
Female tarsometatarsus length	+	
Laying date		
Egg length		+
Egg width		
Average temperature	+	+
Daily temperature difference		
Daily maximum temperature		
Rainfall		
Population		
*AICc*	−197.50	−197.00
Weight	0.948	0.611

+ indicates that the selected parameters of the optimal model; Weight was the proportion of the selected parameters in all parameters of the optimal model.

## Data Availability

The data presented in this study are available on request from the corresponding author.

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
