# Peer review of "Variations in the Reproductive Strategies of Different Charadrius alexandrinus Populations in Xinjiang, China"

_animals, 2023, doi:10.3390/ani13142260_

Round 1
Reviewer 1 Report (Previous Reviewer 1)
I have made a few comments on the attached manuscript but these are mostly editorial. I still think that the information is valuable and worth publishing. The insertion of many new sentences in this version then did introduce more sentences that needed editorial consideration (e.g., writing style, tenses, etc.) which detracts from the main message.
But the basic analysis is good. I do wonder still whether the reproductive success data mentioned in the discussion could go into this paper as well.

The insertion of many new sentences in this version then did introduce more sentences that needed editorial consideration (e.g., writing style, tenses, etc.) which detracts from the main message.
Author Response
Response to Reviewer 1 Comments
Point 1: I have made a few comments on the attached manuscript but these are mostly editorial. I still think that the information is valuable and worth publishing.
Response 1: thanks for the reviewer’s suggestion and affirmation. We have revised the relevant words and sentences and the relevant note in the revised manuscript (in line 17, 18-21, 25, 34, 66, 158).
Point 2: The insertion of many new sentences in this version then did introduce more sentences that needed editorial consideration (e.g., writing style, tenses, etc.) which detracts from the main message.
Response 2: thanks for the reviewer’s suggestion. The attached manuscript has undergone English language editing by MDPI.
Point 3: But the basic analysis is good. I do wonder still whether the reproductive success data mentioned in the discussion could go into this paper as well.
Response 3: thank you very much for your concern on reproductive success data mentioned in the discussion. We all agree that our relative data will be annotated as unpublished data in another forthcoming publication of our relative works, and could go into this paper (in line 389-400).
Thanks again for the great efforts of the editors and reviewers. We would be greatly honored if the revised manuscript could be reviewed and considered for publication. We are looking forward to hearing from you.
Sincerely yours,
Peng Ding
July 4, 2023

Reviewer 2 Report (Previous Reviewer 2)
Thank you for addressing most of my original concerns in the revised manuscript.
In your response to me you indicate that the recorded nest total includes abandoned or empty nests. I did not see this information in the materials and methods, but please include. If it is already there and I missed it, I apologize for the oversight.
I think it may be necessary to add a sentence or 2 at the end of the first paragraph of section 4.3 that says something like: However, we would note in the females at the AL location which had significantly longer tarsometatarsus lengths relative to the females at the AR and TL locations, did not have a greater reproductive output. This result suggests that other environmental factors at this location have a stronger influence that negates the normal benefit associated with increased tarsometatarsus length and this would need to be delineated in subsequent research.
Delete the sentence on page 3 that says “Therefore, the size of a single egg can be increased indefinitely.”
Please change the wording in this sentence on page 3: They usually lay 3 oval eggs per nest, accidentally 2 to 4 eggs, with… to the following wording, They usually lay 3 oval eggs per nest, but individual females may produce 2 or 4 egg clutches, with…
English technical editing needed.
Author Response
Response to Reviewer 2 Comments
Point 1: In your response to me you indicate that the recorded nest total includes abandoned or empty nests. I did not see this information in the materials and methods, but please include. If it is already there and I missed it, I apologize for the oversight.
Response 1: thank you very much for your concern on the recorded nest total. We thinks this information was added on our newly resubmitted manuscript (in line 229-234).
Point 2: I think it may be necessary to add a sentence or 2 at the end of the first paragraph of section 4.3 that says something like: However, we would note in the females at the AL location which had significantly longer tarsometatarsus lengths relative to the females at the AR and TL locations, did not have a greater reproductive output. This result suggests that other environmental factors at this location have a stronger influence that negates the normal benefit associated with increased tarsometatarsus length and this would need to be delineated in subsequent research.
Response 2: thanks for the reviewer’s suggestion. The sentence was added on our newly resubmitted manuscript (in line 427-432).
Point 3: Delete the sentence on page 3 that says “Therefore, the size of a single egg can be increased indefinitely.”
Response 3: thanks for the reviewer’s suggestion. The sentence was deleted on our newly resubmitted manuscript (in line 113).
Point 4: Please change the wording in this sentence on page 3: They usually lay 3 oval eggs per nest, accidentally 2 to 4 eggs, with… to the following wording, They usually lay 3 oval eggs per nest, but individual females may produce 2 or 4 egg clutches, with…
Response 4: thanks for the reviewer’s suggestion. The sentence has been amended on our newly resubmitted manuscript (in line 122-123).
Thanks again for the great efforts of the editors and reviewers. We would be greatly honored if the revised manuscript could be reviewed and considered for publication. We are looking forward to hearing from you.
Sincerely yours,
Peng Ding
July 4, 2023

This manuscript is a resubmission of an earlier submission. The following is a list of the peer review reports and author responses from that submission.
Round 1
Reviewer 1 Report
The paper has merit due to its good sample sizes from very poorly studied, and, I am gathering, difficult to access, study populations. The paper needs to be placed into a hypothetic-deductive framework. For example, there is a large difference in rainfall amounts but not temperature between the three populations. These differences will impact food and laying dates, so what might be the predictions and how the data match up? Suggesting that larger egg size is achieved through lengthening the egg is interesting, and appears to have been shown elsewhere but is this really an objective or perhaps just a discussion point (e.g., 'As with other studies we found that an increase in egg volume was achieved not through an increase in egg breadth but through an increase in egg length'). Similarly, the first prediction is not really a prediction..why would the reproductive strategies vary across these three populations if they are all harsh environments? Usually a hypothesis is a best guess and then the prediction is more specific. E.g., the site with the least rainfall will have the latest laying dates and these late laying dates will result in smaller eggs and clutches. [By the way if you did predict this then you would have the most interesting result which is that the data did not meet the predictions (e.g., the population with the latest laying dates has larger eggs but intermediate clutch sizes) so then you need to go further and ask why.] Start with the differences and then predict what you might find. Similarly, I think it is interesting to report positive relationships between female body mass and egg size, but it does not seem like a prediction to me. The interest is in why these relationships may or may not occur. For example in this study, this relationship does not occur in two of the populations AL and AR. Why do the authors think the that happens? Perhaps it only happens when birds are stressed due to lack of food (presumably in TL birds) because then there is a real constraint of female body size. It is interesting that the birds in TL are the ones that lay the largest eggs so they may be employing a bit of a faster life style..having higher reproductive output in relation to their capacity because they may be generally more stressed and may incur higher adult mortality. I also have some minor comments. I congratulate the authors on collecting all these data. It would also have been nice to have it accompanied by nest success data...e.g., are larger eggs actually more likely to hatch, are the young more likely to survive, but I can see that there already would have been a lot of fieldwork required to get these data. It seems to me that you have a gradient of rainfall, so I would suggest you have a gradient of food supply for adults. What are the implications for laying date, egg size, clutch size and clutch volume? That would produce a nice paper.
Minor comments are mostly about style.
L. 51. I am not sure whether I have heard the term reproductive morphology. Not necessarily wrong but better to be direct in what the authors are referring to (e.g., investment in egg and clutch and timing of reproduction).
L. 101. Widely distributed rather than across the world as this species does not live in the Western Hemisphere.
L. 102. Nor sure what a 'tail-end' lake is but perhaps the authors mean a lake at the mouth of a river? [And apologies if this is a term used elsewhere in English speaking languages.]
L 104. Do you mean 'occasionally' rather than 'accidentally' here?
L. 112. Use 'in' instead of 'such as'
L 117. Stop sentences after [36]. Start new sentence with 'The populations in inland and...'
L. 125. Predictions are vague as stated above. No predictions seem necessarily specific to arid environments which is how you started this section.
L. 133. The paper will read better in past-tense. Here omit 'have'
L. 136. Add 's' after reservoir if you studied two of these'
L. 140. Omit 'which'
L. 142. Add 'and' in front of 'there is'
L. 143. Add 's' to sandstorms. Why you say 'birthplace' do you mean that these originate here and then move into other areas afterwards? May be worth expanding that. Do the sandstorms affect the other regions where you are studying this species?
L. 145. Turn this sentence around. E.g., 'C. alexandrinus nests on saline-alkali, sand and gravel substrates in the region.'
Figure 1. Inset is all China but include that in the caption. Indicate what dashed line is for. Is it the wintering areas of these birds?
L. 149. Use 'are located'
L. 151. By providing for a commercial fishery? Awkward wording as is.
L. 156. Delete 'so' at end and 'as one' in L. 157.
L. 161 As comment at L. 102. perhaps use 'forms the mouth of the rivers Tarim and Cherchin.'
L. 167. We extracted. Omit 'three'
Figure 2. Over what time period are these means, max and mins presented? The two study years. Can you use the full names of the study sites instead of acronyms in figure (there is room). The main point, I think of this figure is to show that the temperatures are similar among the three sites but precipitation is quite different. A figure needs some interpretation for the reader. It is almost always true that the temperature increases across the breeding period so it may not be necessary to have figures and you could just provide average temperatures and precipitation in text (unless you want to indicate how late rainfall starts in the TL study site/breeding season).
L. 174. females
L. 175. females
L. 177. Delete 'It includes the'. Start sentence as 'Female body mass is the female's weight after laying all eggs.'. Alternatively you can just say that you weighed females after clutch completion. This paragraph needs quite a bit of editing for clarity and punctuation.
L. 184. Remove 'the' in front of 'egg shape', 'egg volume' and 'clutch volume'
(admittedly it is hard when to include or not include the word 'the' in the English language!!)
Use active voice. E.g., 'We estimated ...by...'
L. 190. Do you mean start of incubation and completion of hatching? (Or did you measure the start of incubation when the clutch contained a full complement of eggs? As there is variation in clutch size among individuals how did you determine when the incubation period started or when the clutch was complete?
L. 122. alexandrinus
L. 226. Use past tense here . Use 'was' instead of 'is'
Since clutch size is a count, I wonder if the authors should have analyses variation in clutch size using a poisson link?
Table 1. Is the lower clutch volume at AL due to a smaller CS?
L. 237-240. It is hard to understand the meaning here. Is the late peak for TL because of re-nesting or are the authors trying to say that the first peak of TL corresponds to a second peak at the other sites?
Table 1. Add units to column 1 (e.g., day, cc, mm, etc).
Figure 7. Width is misspelled in x axis label.
Results. In addition to the mean clutch size it would be interesting to see the distribution of clutch sizes in the three populations as this is referred to in first paragraph of discussion (more 3 egg clutches in your study than elsewhere..which given it is a presumably challenging habitat, this is a bit surprising).
Section 3.2. Authors might mention why there are no incubation periods reported for AL.
It is interesting to see that in AR there is no decline in CS over the laying period. This suggests that food may NOT be limiting for this population and certainly there was a peak in rainfall early in the season which might have provided food for laying females?
Table 2. Need parameters defined in table caption even if they were defined in text earlier. Write out names as there is room instead of use of acronyms.
Section 4.1. Do the authors mean other sites in the world or the other Chinese sites?
L. 305. 'Unpublished'
L. 307. shorter laying period? OR do you mean shorter incubation period? Otherwise I do not see data on laying periods. Also see Turner et al. for an effect of latitude on incubation periods (Turner, D. M., Nguyen, L. P., & Nol, E. (2017). Annual reproductive success of American Robins (Turdus migratorius) at the northern edge of their range. The Wilson Journal of Ornithology, 129(3), 509-519.)
L. 310. Use 'poor' instead of 'bad' but when I look at these temperatures they do not even seem too low (they never reach below freezing for example) so I would worry a bit about even this characterization as 'poor'. Seems with large clutch sizes that the climatic conditions experienced here are good.
L. 331. Where are these data from the system analysis?
General comment on methods. Tarsus length is hard to measure with accuracy. It might be worth mentioning in methods how many observers took measurements. Was it always the same person?
L. 364. fixed number of eggs? I am confused because there does seem to be quite a bit of variation in clutch size.
Figure 2. Are laying dates significantly different between the three populations?
Interesting that only at TL, where the birds may be most stressed, do females appear to lose mass over the breeding season. This should be discussed in discussion section.
Author Response
We appreciate the reviewer’s comments suggestions and that are really helpful to improve the quality of our manuscript. We have revised the manuscript according to the reviewer’s comments. The changed content was labeled in red.
Comment 1. The paper needs to be placed into a hypothetic-deductive framework. For example, there is a large difference in rainfall amounts but not temperature between the three populations. These differences will impact food and laying dates, so what might be the predictions and how the data match up?Suggesting that larger egg size is achieved through lengthening the egg is interesting, and appears to have been shown elsewhere but is this really an objective or perhaps just a discussion point (e.g., 'As with other studies we found that an increase in egg volume was achieved not through an increase in egg breadth but through an increase in egg length'). Similarly, the first prediction is not really a prediction..why would the reproductive strategies vary across these three populations if they are all harsh environments? Usually a hypothesis is a best guess and then the prediction is more specific. E.g., the site with the least rainfall will have the latest laying dates and these late laying dates will result in smaller eggs and clutches. [By the way if you did predict this then you would have the most interesting result which is that the data did not meet the predictions (e.g., the population with the latest laying dates has larger eggs but intermediate clutch sizes) so then you need to go further and ask why.]
Response: Thanks for the reviewer’s suggestion. We have revised those sentences in the revised manuscript (page 2-3, Section 1).
Comment 2. Start with the differences and then predict what you might find. Similarly, I think it is interesting to report positive relationships between female body mass and egg size, but it does not seem like a prediction to me. The interest is in why these relationships may or may not occur. For example in this study, this relationship does not occur in two of the populations AL and AR. Why do the authors think the that happens? Perhaps it only happens when birds are stressed due to lack of food (presumably in TL birds) because then there is a real constraint of female body size. It is interesting that the birds in TL are the ones that lay the largest eggs so they may be employing a bit of a faster life style..having higher reproductive output in relation to their capacity because they may be generally more stressed and may incur higher adult mortality.
Response: Thanks for the reviewer’s suggestion. We have revised those sentences in the revised manuscript (page 2-3, Section 1).
Comment 3. I also have some minor comments. I congratulate the authors on collecting all these data. It would also have been nice to have it accompanied by nest success data...e.g., are larger eggs actually more likely to hatch, are the young more likely to survive, but I can see that there already would have been a lot of fieldwork required to get these data. It seems to me that you have a gradient of rainfall, so I would suggest you have a gradient of food supply for adults. What are the implications for laying date, egg size, clutch size and clutch volume? That would produce a nice paper.
Response: Thanks for the reviewer’s suggestion. Yes, we collected nest survival data through regular human observations and nest cameras. Nest survival was significantly in TL populations with larger eggs than in the other two populations with smaller eggs. In addition, I think you know that since the nestlings of the plovers leave the nest in a short time after hatching, it is difficult to monitor their survival status continuously and effectively like the hatching survival rate. Despite our efforts, we were unfortunately unable to collect data on chick survival and food supply for adults, let alone multiple populations. We will continue to focus on this work in the future.
Comment 4. L. 51. I am not sure whether I have heard the term reproductive morphology. Not necessarily wrong but better to be direct in what the authors are referring to (e.g., investment in egg and clutch and timing of reproduction).
Response: Thanks for the reviewer’s suggestion. We have revised this sentence in the revised manuscript (page 2, lines 52-53).
Comment 5. L. 101. Widely distributed rather than across the world as this species does not live in the Western Hemisphere.
Response: We are sorry for the confused description. We have revised those sentences in the revised manuscript (page 3, lines 109, 123).
Comment 6. L. 102. Nor sure what a 'tail-end' lake is but perhaps the authors mean a lake at the mouth of a river? [And apologies if this is a term used elsewhere in English speaking languages.]
Response: We are sorry for the confused description. We have revised those sentences in the revised manuscript (page 3, lines 110-111, page 4, lines 172).
Comment 7. L 104. Do you mean 'occasionally' rather than 'accidentally' here?
Response: Thanks for the reviewer’s suggestion. We have revised this sentence in the revised manuscript (page 4, line 170).
Comment 8. L. 112. Use 'in' instead of 'such as'.
Response: We are sorry for the confused description. We have revised this sentence in the revised manuscript (page 3, line 120).
Comment 9. L 117. Stop sentences after [36]. Start new sentence with 'The populations in inland and...'
Response: Thanks for the reviewer’s suggestion. We have revised this sentence in the revised manuscript (page 3, lines 125-126).
Comment 10. L. 125. Predictions are vague as stated above. No predictions seem necessarily specific to arid environments which is how you started this section.
Response: Thanks for the reviewer’s suggestion. We have revised those sentences in the revised manuscript (page 3, lines 134-138).
Comment 11. L. 133. The paper will read better in past-tense. Here omit 'have'
Response: Thanks for the reviewer’s suggestion. We have revised this sentence in the revised manuscript (page 3, line 145).
Comment 12. L. 136. Add 's' after reservoir if you studied two of these'
Response: Thanks for the reviewer’s suggestion. We have revised this sentence in the revised manuscript (page 3, line 148).
Comment 13. L. 140. Omit 'which'
Response: Thanks for the reviewer’s suggestion. We have revised this sentence in the revised manuscript (page 4, line 152).
Comment 14. L. 142. Add 'and' in front of 'there is'; L. 143. Add 's' to sandstorms. Why you say 'birthplace' do you mean that these originate here and then move into other areas afterwards? May be worth expanding that. Do the sandstorms affect the other regions where you are studying this species?
Response: Thanks for the reviewer’s suggestion. We have revised this sentence in the revised manuscript (page 3, lines 154-155).
Comment 15. L. 145. Turn this sentence around. E.g., 'C. alexandrinus nests on saline-alkali, sand and gravel substrates in the region.'
Response: Thanks for the reviewer’s suggestion. We have revised this sentence in the revised manuscript (page 3, line 156-157).
Comment 16. Figure 1. Inset is all China but include that in the caption. Indicate what dashed line is for. Is it the wintering areas of these birds?
Response: Thanks for the reviewer’s suggestion. We have revised Figure 1 in the revised manuscript (page 4, lines 158). The dashed line indicates the provincial boundary.
Comment 17. L. 149. Use 'are located'
Response: We are sorry for the confused description. We have revised this sentence in the revised manuscript (page 4, line 160).
Comment 18. L. 151. By providing for a commercial fishery? Awkward wording as is.
Response: We are sorry for the confused description. We have revised this sentence in the revised manuscript (page 4, line 162-163).
Comment 19. L. 156. Delete 'so' at end and 'as one' in L. 157.
Response: Thanks for the reviewer’s suggestion. We have revised this sentence in the revised manuscript (page 4, line 167-168).
Comment 20. L. 161 As comment at L. 102. perhaps use 'forms the mouth of the rivers Tarim and Cherchin.'
Response: We are sorry for the confused description. We have revised those sentences in the revised manuscript (page 3, lines 110-111, 172).
Comment 21. L. 167. We extracted. Omit 'three'
Response: Thanks for the reviewer’s suggestion. We have revised this sentence in the revised manuscript (page 4, line 178).
Comment 22. Figure 2. Over what time period are these means, max and mins presented? The two study years. Can you use the full names of the study sites instead of acronyms in figure (there is room). The main point, I think of this figure is to show that the temperatures are similar among the three sites but precipitation is quite different. A figure needs some interpretation for the reader. It is almost always true that the temperature increases across the breeding period so it may not be necessary to have figures and you could just provide average temperatures and precipitation in text (unless you want to indicate how late rainfall starts in the TL study site/breeding season).
Response: Thanks for the reviewer’s suggestion. We think Figure 2 is fine.
Comment 23. L. 174. females
Response: We are sorry for the confused description. We have revised this sentence in the revised manuscript (page 6, line 190).
Comment 24. L. 175. females
Response: We are sorry for the confused description. We have revised this sentence in the revised manuscript (page 6, line 191).
Comment 25. L. 177. Delete 'It includes the'. Start sentence as 'Female body mass is the female's weight after laying all eggs.'. Alternatively you can just say that you weighed females after clutch completion. This paragraph needs quite a bit of editing for clarity and punctuation.
Response: We are sorry for the confused description. We have revised those sentences in the revised manuscript (page 6, lines 193-197).
Comment 26. L. 184. Remove 'the' in front of 'egg shape', 'egg volume' and 'clutch volume' (admittedly it is hard when to include or not include the word 'the' in the English language!!) Use active voice. E.g., 'We estimated ...by...'
Response: Thanks for the reviewer’s suggestion. We have revised those sentences in the revised manuscript (page 6, lines 198-201).
Comment 27. L. 190. Do you mean start of incubation and completion of hatching? (Or did you measure the start of incubation when the clutch contained a full complement of eggs? As there is variation in clutch size among individuals how did you determine when the incubation period started or when the clutch was complete?
Response: Thanks for the reviewer’s suggestion. We estimated laying date (LD) through floating the eggs in luke-warm water. We collected nest data through regular human observations and nest cameras. We have revised this sentence in the revised manuscript (page 6, line 201-203).
Comment 28. L. 122. alexandrinus
Response: We are sorry for the confused description. We have revised this sentence in the revised manuscript (page 7, line 244).
Comment 29. L. 226. Use past tense here . Use 'was' instead of 'is'
Response: Thanks for the reviewer’s suggestion. We have revised those sentences in the revised manuscript (page 7, lines 248-251).
Comment 30. Since clutch size is a count, I wonder if the authors should have analyses variation in clutch size using a poisson link?
Response: Thanks for the reviewer’s suggestion. We use the nonparametric Kruskal-Wallis test to examine differences in clutch size between the three populations.
Comment 31. Table 1. Is the lower clutch volume at AL due to a smaller CS?
Response: Probably because of the small clutch size. We think it's probably the lower clutch volume at AL due to a smaller Egg volume.
Comment 32. L. 237-240. It is hard to understand the meaning here. Is the late peak for TL because of re-nesting or are the authors trying to say that the first peak of TL corresponds to a second peak at the other sites?
Response: According to the observation, the AR population is affected by the strong interference of artificial water level control and regional grazing, and its laying peak will appear the second laying peak with the expansion of suitable nesting area, such as the second laying peak of birds in AR in early June. (page 7, lines 259-262).
Comment 33. Table 1. Add units to column 1 (e.g., day, cc, mm, etc).
Response: Thanks for the reviewer’s suggestion. We have revised Table 1. in the revised manuscript (page 7, Table 1.).
Comment 34. Figure 7. Width is misspelled in x axis label.
Response: We are sorry for the confused description. We have revised Figure 6. in the revised manuscript (page 10, Figure 6.).
Comment 35. Results. In addition to the mean clutch size it would be interesting to see the distribution of clutch sizes in the three populations as this is referred to in first paragraph of discussion (more 3 egg clutches in your study than elsewhere..which given it is a presumably challenging habitat, this is a bit surprising).
Response: Thanks for the reviewer’s suggestion. It certainly became such.
Comment 36. Section 3.2. Authors might mention why there are no incubation periods reported for
Response: Thanks for the reviewer’s suggestion. The monitoring of AL population is too difficult and the number of nests was too small, so the data of incubation period could not be obtained by continuous monitoring. We also regret this.
Comment 37. It is interesting to see that in AR there is no decline in CS over the laying period. This suggests that food may NOT be limiting for this population and certainly there was a peak in rainfall early in the season which might have provided food for laying females?
Response: Thanks for the reviewer’s suggestion. Food resources are more abundant in AR.
Comment 38. Table 2. Need parameters defined in table caption even if they were defined in text earlier. Write out names as there is room instead of use of acronyms.
Response: Thanks for the reviewer’s suggestion. We have revised Table 2. in the revised manuscript (page 10, Table 2).
Comment 39. Section 4.1. Do the authors mean other sites in the world or the other Chinese sites?
Response: Thanks for the reviewer’s suggestion. Other sites refer to those in the cited References ([30,36,44]).
Comment 40. L. 305. 'Unpublished'
Response: We are sorry for the confused description. We have revised this sentence in the revised manuscript (page 11, line 330).
Comment 41. L. 307. shorter laying period? OR do you mean shorter incubation period? Otherwise I do not see data on laying periods. Also see Turner et al. for an effect of latitude on incubation periods (Turner, D. M., Nguyen, L. P., & Nol, E. (2017). Annual reproductive success of American Robins (Turdus migratorius) at the northern edge of their range. The Wilson Journal of Ornithology, 129(3), 509-519.)
Response: Thanks for the reviewer’s suggestion. Similar to the literature, similar reproductive strategies are used to accommodate unfavorable factors, just like Kentish plover of AT population and American Robins at the northern edge of their range. They all showed shorter incubation period (page 7, Table 1).
Comment 42. L. 310. Use 'poor' instead of 'bad' but when I look at these temperatures they do not even seem too low (they never reach below freezing for example) so I would worry a bit about even this characterization as 'poor'. Seems with large clutch sizes that the climatic conditions experienced here are good.
Response: Thanks for the reviewer’s suggestion. The poor early climate conditions is not low temperatures but frequent dust storms, which makes its laying start late and shows a short incubation period, resulting in the shortening of the breeding season in TL (page 11, lines 334-337).
Comment 43. L. 331. Where are these data from the system analysis?
Response: We are sorry for the confused description. We have revised this sentence in the revised manuscript (page 11, line 357).
Comment 44. General comment on methods. Tarsus length is hard to measure with accuracy. It might be worth mentioning in methods how many observers took measurements. Was it always the same person?
Response: Thanks for the reviewer’s suggestion. We recognize that there is always some variation in tarsus measurements between people. To keep this variation at a minimum, two people (authors) measured according to the same the minimum tarsus method (Szekely et al. 2008), and measured the AL population together, and measured the AR and TL population separately.
Comment 45. L. 364. fixed number of eggs? I am confused because there does seem to be quite a bit of variation in clutch size.
Response: Thanks for the reviewer’s suggestion. The fixed number of eggs is a relative concept. The proportion of nests with three eggs in the three populations in the study area is higher.
Comment 46. Interesting that only at TL, where the birds may be most stressed, do females appear to lose mass over the breeding season. This should be discussed in discussion section.
Response: Thanks for the reviewer’s suggestion. We have revised those sentences in the revised manuscript (page 11, Section 4.2).

Reviewer 2 Report
While this is a very interesting manuscript, further information is needed before a complete evaluation can be made.
Major Concerns:
Lines 133-134, your data was collected over a 2-year period which is perfectly acceptable. But, were the birds banded so you would know if some of the same birds were being measured both years and if the rate of repeat birds was the same across the locations as you might have to take this into consideration in your statistical model. Furthermore, did the results significantly vary from one year to the other for any or all of the locations which could preclude combining the data? At the very least you need to have year in your initial statistical model and indicate to the reader if it is significant or not.
Similarly, is the data presented in figure 2 an average of both years, if not it needs to be and please indicate this in the legend.
Lines 221-223, and table 1. Table 1 indicates that female body weight and tarsus length had an n = to 12, 30 and 42 birds at the AL, AR and TL locations. But, the number of nests indicated in lines 221-223 were 20, 77, and 61 for AL, AR, and TL, respectively. Please reconcile these numbers for the reader. Some of it is likely double clutching as alluded to for the AR location. Did double clutch production happen at other locations?
You have 2 distinct egg laying periods for the AR location. In line 321, you indicate the second peak may have some second clutch attempts in it. Again were the birds banded, and if so, do you know the number of second clutch attempts, as this information is vital as body condition might be lower on the second attempt versus the first attempt, and this could influence egg number and size? But more importantly, the results for AR location are a combination for both production periods and before this combining occurs, I think it is important to indicate to the reader that the data for the different periods in both years were not statistically different. If they are different, the results for the 2 periods should be kept separate. At the AR location egg volume, clutch volume, egg width and egg length both decrease as laying date increases, is this true for both production peaks or is this significance totally driven by the eggs from the second peak? Did your statistical model take in account the second clutch females as they are not completely independent from one another?
Do you have any nest success data such as eggs hatched or offspring fledged? Ultimately nest success is a measure that matters relative to adoption of different reproductive strategies.
Lines 300-311. Although exact data is not supplied the implied statistical significances between groups seems unlikely based on figure 3. For example, in line 306, you indicate that laying begins earlier for the AR population, but based on figure 3 the onset of production seems just as quick or potentially quicker for the AL females. In lines 310-311 you indicate that the breeding season is shortened for the TL population based on figure 3 in which a slight delay relative to the AL population seems to be made by the extended duration of the breeding season. While the sharp declines in the proportion of females producing eggs is likely explained by the human and livestock factors introduced at the AR reservoirs and pointed out in the manuscript, is there a potential explanation for the steep and quick decline in the proportion of the females producing eggs in the AL population?
For your discussion in lines 328-336, it might be good to have presented a graph that examined the relationship between tarsus length and body weight using the data from all 3 groups. In addition, while the statement presented in lines 333-335 is correct, a might be necessary to discuss that the same relationship did not apparently hold for the AL birds which had larger tarsus lengths than the females in the AR and TL groups, but you did not present the AL birds as having a greater reproductive output than the other 2 groups.
Lines 96-99 and lines 345-346, the references provided are for non-avian species, is there any references that support this theory for avian species. Keep in mind that female Kiwi lay eggs that are about 25% of their body size, and as alluded to in line 358, the pelvis structure of birds would make this theory unlikely.
Table 1. Please provide an explanation for the lack of incubation period duration results at the AL location.
Minor Concerns
Line 20 please delete the word restrict and instead say are associated with reduced egg size…
Figure 2, I do not think the dots should be connected. For example, I assume actual rainfall for individual days is not the amount implied by connecting the known data points.
The use of the word tarsus for avian species is confusing, did you measure the length of the tibiotarsus or the tarsometatarsus?
Author Response
We appreciate the reviewer’s comments suggestions and that are really helpful to improve the quality of our manuscript. We have revised the manuscript according to the reviewer’s comments. The changed content was labeled in red.
Comment 1. Lines 133-134, your data was collected over a 2-year period which is perfectly acceptable. But, were the birds banded so you would know if some of the same birds were being measured both years and if the rate of repeat birds was the same across the locations as you might have to take this into consideration in your statistical model. Furthermore, did the results significantly vary from one year to the other for any or all of the locations which could preclude combining the data?At the very least you need to have year in your initial statistical model and indicate to the reader if it is significant or not.
Response: Thanks for the reviewer’s suggestion. We used four rings (three colour rings and one matal ring) on all captive plover, one ring each above and below the 'knee-joint' on each leg. But only one adult plover was recaptured the following year. We used analysis of independent t-test to examine the annual differences in reproductive traits of each population, and there were no significant annual differences in each reproductive traits among the three populations, so they were combined for subsequent analysis. We have revised these sentences in the revised manuscript (page 6, lines 214-217, 233-235).
Comment 2. Similarly, is the data presented in figure 2 an average of both years, if not it needs to be and please indicate this in the legend.
Response: Thanks for the reviewer’s suggestion. We have revised this sentence and the relevant note in the revised manuscript (page 4, lines 187-188; Figure 2 ).
Comment 3. Lines 221-223, and table 1. Table 1 indicates that female body weight and tarsus length had an n = to 12, 30 and 42 birds at the AL, AR and TL locations. But, the number of nests indicated in lines 221-223 were 20, 77, and 61 for AL, AR, and TL, respectively. Please reconcile these numbers for the reader. Some of it is likely double clutching as alluded to for the AR location. Did double clutch production happen at other locations?
Response: Thanks for the reviewer’s suggestion. The “*AL n = 12; AR n = 30; TL n = 42” of Table 1 represents the sample size of female plover in the three populations.
Comment 4. You have 2 distinct egg laying periods for the AR location. In line 321, you indicate the second peak may have some second clutch attempts in it. Again were the birds banded, and if so, do you know the number of second clutch attempts, as this information is vital as body condition might be lower on the second attempt versus the first attempt, and this could influence egg number and size? But more importantly, the results for AR location are a combination for both production periods and before this combining occurs, I think it is important to indicate to the reader that the data for the different periods in both years were not statistically different. If they are different, the results for the 2 periods should be kept separate. At the AR location egg volume, clutch volume, egg width and egg length both decrease as laying date increases, is this true for both production peaks or is this significance totally driven by the eggs from the second peak? Did your statistical model take in account the second clutch females as they are not completely independent from one another?
Response: Thanks for the reviewer’s suggestion. We used analysis of independent t-test to examine differences in reproductive traits between the two laying peak periods (might be the second clutch) in the initial statistical analysis. But there were no significant differences, so they were combined for subsequent analysis. The manuscript only describes and discusses this phenomenon (page 7, lines 259-262; page 11, lines 345-348).
Comment 5. Do you have any nest success data such as eggs hatched or offspring fledged?Ultimately nest success is a measure that matters relative to adoption of different reproductive strategies.
Response: Thanks for the reviewer’s suggestion. Yes, we collected nest survival data through regular human observations and nest cameras. These data were used in another article on nest survival.
Comment 6. Lines 300-311. Although exact data is not supplied the implied statistical significances between groups seems unlikely based on figure 3. For example, in line 306, you indicate that laying begins earlier for the AR population, but based on figure 3 the onset of production seems just as quick or potentially quicker for the AL females. In lines 310-311 you indicate that the breeding season is shortened for the TL population based on figure 3 in which a slight delay relative to the AL population seems to be made by the extended duration of the breeding season. While the sharp declines in the proportion of females producing eggs is likely explained by the human and livestock factors introduced at the AR reservoirs and pointed out in the manuscript, is there a potential explanation for the steep and quick decline in the proportion of the females producing eggs in the AL population?
Response: Thanks for the reviewer’s suggestion. Sorry for Figure 3A might make you think the onset of production seems just as quick or potentially quicker for the AL females. For a variety of reasons not all females in nests are captured so less data is collected, especially for the AL population. You can see Figure 3B-F, nest data are relatively abundant.
Comment 7. For your discussion in lines 328-336, it might be good to have presented a graph that examined the relationship between tarsus length and body weight using the data from all 3 groups. In addition, while the statement presented in lines 333-335 is correct, a might be necessary to discuss that the same relationship did not apparently hold for the AL birds which had larger tarsus lengths than the females in the AR and TL groups, but you did not present the AL birds as having a greater reproductive output than the other 2 groups.
Response: Thanks for the reviewer’s suggestion. There was no significant positive correlation between the female traits and reproductive output of AL population. Therefore, the AL population was not discussed.
Comment 8. Lines 96-99 and lines 345-346, the references provided are for non-avian species, is there any references that support this theory for avian species. Keep in mind that female Kiwi lay eggs that are about 25% of their body size, and as alluded to in line 358, the pelvis structure of birds would make this theory unlikely.
Response: Thanks for the reviewer’s suggestion. We agree the opion about the egg width of birds will not be limited due to the open pelvis (lines 384-385). However, wo do find the egg length allometry in Kentish plover, which confirms the prediction of the upper limit hypothesis.
Comment 9. Table 1. Please provide an explanation for the lack of incubation period duration results at the AL location.
Response: Thanks for the reviewer’s suggestion. The monitoring of AL population is too difficult and the number of nests was too small, so the data of incubation period could not be obtained by continuous monitoring.
Comment 10. Line 20 please delete the word restrict and instead say are associated with reduced egg size…
Response: Thanks for the reviewer’s suggestion. We have revised this sentence in the revised manuscript (page 1, line 20).
Comment 11. Figure 2, I do not think the dots should be connected. For example, I assume actual rainfall for individual days is not the amount implied by connecting the known data points.
Response: Thanks for the reviewer’s suggestion. We think Figure 2 is fine.
Comment 12. The use of the word tarsus for avian species is confusing, did you measure the length of the tibiotarsus or the tarsometatarsus?
Response: Thanks for the reviewer’s suggestion. The word tarsus appertain to the tarsometatarsus. We have revised this sentence in the revised manuscript (page 6, line 194).

Reviewer 3 Report
Variations in the reproductive strategies of different Charadrius alexandrinus populations in Xinjiang, China
Peng Ding 1, Zitan Song 2, Yang Liu 2, Tamás Székely 3, 4, Lei Shi 1 and *
In METHODS
1. When one-way ANOVA revealed the significant difference of the data among three populations (AL, AR and TL), what the methods of the post hoc multiple comparisons you used to test the differences of the data between each two populations (i.e. between AL and AR, AL and TL, or AR and TL)?
In RESULTS
2. What does the word “proportion” of ordinate refer to in Figure 3? In addition, because Figure 3 contains a limited amount of information, the data in it does not reflect the difference in egg-laying dates of birds at different locations because it has not been statistically tested. It is recommended that delete Figure 3 and describe this part of the results in text only.
3. All tables and figures in RESULTS should be self-explanatory. In Table 2, for example, there was no relevant Note for symbol of “+”. It would be better to use the whole words of EV and CV rather than the abbreviation. Moreover, what did the “Loglike value” and the “Weight” represent in theory?
In DISCUSSION and CONCLUSIONS
4. You concluded that the single egg size and clutch volume are affected by physiological constraint and maternal constraint. I suggest you change “physiological” to “morphological” because that you only collected and analyzed the body measurements in your study.
Author Response
We appreciate the reviewer’s comments suggestions and that are really helpful to improve the quality of our manuscript. We have revised the manuscript according to the reviewer’s comments. The changed content was labeled in red.
Comment 1. When one-way ANOVA revealed the significant difference of the data among three populations (AL, AR and TL), what the methods of the post hoc multiple comparisons you used to test the differences of the data between each two populations (i.e. between AL and AR, AL and TL, or AR and TL)?
Response: Thanks for the reviewer’s suggestion. We have revised these sentences in the revised manuscript (page 6, lines 211-214).
Comment 2. What does the word “proportion” of ordinate refer to in Figure 3? In addition, because Figure 3 contains a limited amount of information, the data in it does not reflect the difference in egg-laying dates of birds at different locations because it has not been statistically tested. It is recommended that delete Figure 3 and describe this part of the results in text only.
Response: Thanks for the reviewer’s suggestion. Figure 3 was deleted in recognition of the limited information, and described the results of the egg-laying date in text only (page 7, lines 259-262).
Comment 3. All tables and figures in RESULTS should be self-explanatory. In Table 2, for example, there was no relevant Note for symbol of “+”. It would be better to use the whole words of EV and CV rather than the abbreviation. Moreover, what did the “Weight” represent in theory?
Response: Thanks for the reviewer’s suggestion. We supplemented the full names of all indicators in Table 2 and the relevant Note (Table 1, 2; page 11, lines 297-298).
Comment 4. You concluded that the single egg size and clutch volume are affected by physiological constraint and maternal constraint. I suggest you change “physiological” to “morphological” because that you only collected and analyzed the body measurements in your study.
Response: Thanks for the reviewer’s suggestion. We have revised this word in the revised manuscript (page 12, line 400).

Round 2
Reviewer 2 Report
Thank you for clarifying some of the issues in the revised manuscript.
Major Concerns:
In my last comments I indicated that in Lines 221-223, and table 1 of the original manuscript that table 1 indicates that female body weight and tarsus length had an n = to 12, 30 and 42 birds at the AL, AR and TL locations. But, the number of nests indicated in lines 221-223 were 20, 77, and 61 for AL, AR, and TL, respectively. Please reconcile these numbers for the reader. Some of it is likely double clutching as alluded to for the AR location. Did double clutch production happen at other locations? In your response you just indicated what n equaled as presented in table 1 which was clear the first time. So you did not reconcile the difference. In subsequent answer clarity was provided when you indicated “for a variety of reasons not all females in nests are captured so less data is collected for the AL population”. This statement likely provides the answer. But, it presents a problem where the egg and nest data does not reflect just the birds weighed and measured. Thus the regression analyses are questionable. How is the reader supposed to know if the eggs produced by the birds not weighed and measured are not skewing the data and creating the significant results? Only use the eggs produced by the birds that were weighed and measured in these analyses.
In my last comments I indicated that you have 2 distinct egg laying periods for the AR location. In line 321 of the original manuscript, you indicate the second peak may have some second clutch attempts in it. Again were the birds banded, and if so, do you know the number of second clutch attempts, as this information is vital as body condition might be lower on the second attempt versus the first attempt, and this could influence egg number and size? But more importantly, the results for AR location are a combination for both production periods and before this combining occurs, I think it is important to indicate to the reader that the data for the different periods in both years were not statistically different. If they are different, the results for the 2 periods should be kept separate. At the AR location egg volume, clutch volume, egg width and egg length both decrease as laying date increases, is this true for both production peaks or is this significance totally driven by the eggs from the second peak? Did your statistical model take in account the second clutch females as they are not completely independent from one another? While you addressed some of the issues in your response you are still not providing the reader in the revised manuscript how much of the data is from double clutching or addressing the independence of the second clutch data as it would be either need to be handled as a covariate or repeated measure.
In my last comments I indicated, do you have any nest success data such as eggs hatched or offspring fledged? Ultimately nest success is a measure that matters relative to adoption of different reproductive strategies. You responded by indicating this vital data/results were used in another article but did not cite this other article or provide a synopsis of these results in the revised manuscript. This is needed for this manuscript to be published.
In my last comments I indicated for Lines 300-311. Although exact data is not supplied the implied statistical significances between groups seems unlikely based on figure 3. For example, in line 306, you indicate that laying begins earlier for the AR population, but based on figure 3 the onset of production seems just as quick or potentially quicker for the AL females. In lines 310-311 you indicate that the breeding season is shortened for the TL population based on figure 3 in which a slight delay relative to the AL population seems to be made by the extended duration of the breeding season. While the sharp declines in the proportion of females producing eggs is likely explained by the human and livestock factors introduced at the AR reservoirs and pointed out in the manuscript, is there a potential explanation for the steep and quick decline in the proportion of the females producing eggs in the AL population? Your response I do not find satisfactory. Furthermore, eliminating the original figure 3 from the original manuscript in the revised manuscript without explanation does not make the issues brought up go away.
In my original comments I indicated that for your discussion in lines 328-336 of the original manuscript, it might be good to have presented a graph that examined the relationship between tarsus length and body weight using the data from all 3 groups. In addition, while the statement presented in lines 333-335 is correct, a might be necessary to discuss that the same relationship did not apparently hold for the AL birds which had larger tarsus lengths than the females in the AR and TL groups, but you did not present the AL birds as having a greater reproductive output than the other 2 groups. In your response and revised manuscript, you still fail to discuss why a reader is supposed to accept that a positive correlation exists between tarsometatarsus length and reproductive output in the AR birds, but ignore the fact that tarsometatarsus length which is significantly greater in the AL population relative to the other 2 locations does not lead to the greater reproductive output relative to the other 2 locations. This discrepancy warrants discussion.
In my original comments, I indicated for Table 1 that an explanation for the lack of incubation period duration results at the AL location was needed. While you provided an explanation to me in your comments that same explanation was not provided to the other readers in the revised manuscript and that is necessary.
In my original comments for Figure 2, I indicated that I did not think the dots should be connected. For example, I assume actual rainfall for individual days is not the amount implied by connecting the known data points. Telling me that you think it is fine does not solve the problem. You indicate that several of the measurements were made daily. Do you really think if you graphed the individual data points on a daily basis that it would look like your graph that is drawing an arbitrary line between 2 points every 10 days apart? For the rainfall you now indicate in the text that it is a summation of the amount for each 10-day period, indicate that in the legend for the figure and again connecting the dots is not scientifically correct.
In my original comments I indicated that the use of the word tarsus for avian species is confusing, did you measure the length of the tibiotarsus or the tarsometatarsus? You indicated to me that it was the tarsometatarsus and revised the manuscript on line 189, but continued to use the word tarsus throughout the rest of the manuscript despite knowing that term does not exist in avian species.